# Current Treatment Options for Metastatic Hormone-Sensitive Prostate Cancer

**DOI:** 10.3390/cancers11091355

**Published:** 2019-09-12

**Authors:** Carlo Cattrini, Elena Castro, Rebeca Lozano, Elisa Zanardi, Alessandra Rubagotti, Francesco Boccardo, David Olmos

**Affiliations:** 1Academic Unit of Medical Oncology, IRCCS San Martino Polyclinic Hospital, 16132 Genoa, Italy; carlo.cattrini@gmail.com (C.C.); elisa.zanardi@unige.it (E.Z.); alessandra.rubagotti@unige.it (A.R.); 2Department of Internal Medicine and Medical Specialties (DIMI), University of Genoa, 16132 Genoa, Italy; 3Prostate Cancer Clinical Research Unit, Spanish National Cancer Research Centre (CNIO), 28029 Madrid, Spain; ecastro@ext.cnio.es (E.C.); rlozano@ext.cnio.es (R.L.); dolmos@cnio.es (D.O.); 4CNIO-IBIMA Genitourinary Cancer Unit, Hospitales Universitarios Virgen de la Victoria y Regional de Málaga, Instituto de Investigación Biomédica de Málaga, 29010 Malaga, Spain; 5Department of Health Sciences (DISSAL), University of Genoa, 16132 Genoa, Italy

**Keywords:** hormone-sensitive prostate cancer, hormone-naïve prostate cancer, docetaxel, enzalutamide, abiraterone acetate, apalutamide, radiotherapy

## Abstract

The possible treatments options for metastatic hormone-sensitive prostate cancer (mHSPC) have dramatically increased during the last years. The old backbone, which androgen-deprivation therapy (ADT) is the exclusive approach for hormone-naïve patients, has been disrupted. Despite the fact that several high-quality, randomized, controlled phase 3 trials have been conducted in this setting, no direct comparison is currently available among the different strategies. Inadequate power, absence of preplanning and small sample size frequently affect the subgroup analyses according to disease volume or patient’s risk. The choice between ADT alone and ADT combined with docetaxel, abiraterone acetate, enzalutamide, apalutamide or radiotherapy to the primary tumor remains challenging. Factors that are related to the tumor, patient or drug side effects, currently guide these clinical decisions. This comprehensive review aims to indirectly compare the phase 3 trials in the mHSPC setting, in order to extrapolate data useful for treatment selection, providing also perspectives on future biomarkers.

## 1. Introduction

Prostate cancer is the most common neoplasm in the Western Countries, and is one of the leading causes of death worldwide [1]. Patients who are treated with prostatectomy or radiotherapy for localized disease often develop metastatic recurrence after local treatment. Some patients can also show de novo metastatic disease without prior radical procedures. Although the timing of metastatic presentation is quite different, all these patients are supposed to be responsive to surgical or medical castration, and are thus affected by a condition that is known as metastatic hormone-naïve, or hormone-sensitive prostate cancer (mHSPC). This disease stage precedes the development of metastatic castration-resistant prostate cancer (mCRPC), which is characterized by poor prognosis and high lethality. 

Androgen-deprivation therapy (ADT) has been the cornerstone of the systemic treatment for mHSPC since the 1940s, when Huggins and Hodges demonstrated the efficacy of hormonal treatment in patients with prostate cancer [2]. 

The addition of first-generation antiandrogens (i.e., bicalutamide) to ADT—a procedure known as complete or maximal androgen blockade (CAB-MAB)—showed a slight survival benefit in mHSPC in the face of numerous side effects [3]. Therefore, the utility of MAB in clinical practice has remained controversial [4]. Intermittent androgen deprivation (IAD) has been investigated in several phase III trials that reached inconclusive and contradictory results [5]. Despite the fact that no randomized trial has ever demonstrated a survival benefit with intermittent compared to continuous ADT, the first approach has been associated with a marked improvement in quality of life (QoL).

In 2015, docetaxel was the first agent to demonstrate a significant survival benefit compared to ADT alone in mHSPC [6]. Subsequently, novel androgen-receptor signaling inhibitors (ARSi) have progressively become new treatment choices for mHSPC, demonstrating significant clinical benefits in multiple endpoints [7]. The lack of predictive biomarkers and the absence of direct comparisons among the different agents are the major current issues to be faced when selecting the best treatment for patients with mHSPC. The present review aims at summarizing the current evidences based on the phase 3 randomized controlled trials, in order to indirectly compare the efficacy and tolerability of the different therapeutic options for mHSPC.

## 2. Prognostic and Predictive Factors in mHSPC

Several prognostic and predictive factors have been proposed in mCRPC [8], whereas less information is available for mHSPC. Metastatic burden and metastasis localization, time of metastatic presentation and the Gleason score are the main prognostic factors that have been identified in clinical trials that included patients with mHSPC. However, it is currently unclear whether the prognostic significance of the Gleason score would be strengthened after the introduction of the new International Society of Urological Pathology (ISUP) classification in 2016, which distinguishes five different Gleason grade groups [9].

### 2.1. Glass and Gravis Models

In 2003, Glass et al. published a prognostic model for mHSPC based on outcomes of patients enrolled in the SWOG 8894 clinical trial [10,11]. This model differentiated three prognostic groups according to four risk factors: localization of bone disease, performance status, PSA levels, and the Gleason score. The good, intermediate, and poor prognosis groups were associated with estimated 5-year survival rates of 42%, 21% and 9%, respectively [10]. Gravis et al. tested the Glass model in a post hoc analysis of the GETUG-AFU 15 cohort, in which 385 mHSPC patients were randomized to ADT with docetaxel, or with just ADT alone [12]. In the GETUG-AFU 15 population, the difference between intermediate and poor prognosis groups was not statistically significant [13], and the overall discriminatory value of the Glass model was low in this population. These findings were attributed to improvements in mHSPC management, widespread PSA screening and differences in patients’ populations between the trials [14]. Gravis et al. later developed and validated a more accurate prognostic model based on the GETUG-15 outcomes [13]. In this model, ALP, the Gleason score and pain intensity showed the greatest degree of discrimination in the recursive partitioning algorithm. However, ALP alone performed as well as the more complex Glass model comprising four risk factors, with similar concordance indices [13]. ALP alone was therefore proposed as a cheap and readily available prognostic factor for patients with mHSPC, but it did not show any predictive value in monitoring the response to upfront docetaxel in the GETUG-15 study [12], further drawing into question its clinical utility [14]. 

### 2.2. Risk Factors in Phase 3 Trials

The CHAARTED and GETUG-AFU 15 trials assessed the role of upfront docetaxel plus ADT versus ADT alone in patients with mHSPC [6,12]. The prospective stratification of high-volume (defined as the presence of visceral metastases or ≥4 bone lesions with ≥1 beyond the vertebral bodies and pelvis) versus low-volume metastatic disease was introduced for the first time as an amendment to the protocol of the CHAARTED trial (Table 1) [6]. Given the uncertain role of chemotherapy in low-volume disease [15], several subsequent trials in mHSPC have analyzed patients’ outcomes based on disease burden (Table 3). Different from the CHAARTED trial, the LATITUDE trial of abiraterone acetate in mHSPC used another classification to define high-risk patients, which was based on the presence of two or more high risk features that included ≥3 bone metastases, visceral metastases and Gleason ≥ 8 (Table 1) [16]. A recent meta-analysis of the aggregate data of patient subgroups from the CHAARTED and GETUG-AFU15 studies evaluated overall survival (OS) according to the metastatic tumor burden and time of metastasis occurrence (at diagnosis or after prior local treatment) [17]. The authors identified three prognostic subgroups: good prognosis for those with prior local treatment and low-volume disease; intermediate prognosis for those with prior local treatment and high-volume disease, or those with low-volume disease and de novo metastases; and poor prognosis for those with de novo high-volume disease. These data were recently confirmed by a retrospective cohort of 436 consecutive patients with mHSPC treated with ADT at the Dana-Farber Cancer Institute between 1990 and 2013 [18]. 

### 2.3. Biomarkers for Treatment Selection

Although few biomarkers are identified in the mHSPC setting, several aberrations that are found in mCRPC are specific of aggressive or metastatic disease, rather than being the result of selective pressure of treatments, and can be found in mHSPC at frequencies that are comparable to mCRPC. For example, the prevalence of alterations in *PTEN*, *TP53*, *FOXA1*, *PIK3A*, *APC* and *BRCA2* did not differ significantly between de novo mHSPC and mCRPC [19,20]. Some of these alterations have been associated with responses to specific treatments. Somatic and germline defects in homologous recombination repair (HRR) have been suggested as potential biomarkers of response to platinum-based chemotherapy [21,22] and poly (ADP-ribose) polymerase (PARP) inhibitors [23,24]. The PROREPAIR-B prospective study has confirmed *BRCA2* as an independent prognostic factor for survival in mCRPC, and this study has also suggested that *BRCA2* might be a predictor of poor response to chemotherapy in a first-line setting [25]. Patients with microsatellite instability or mismatch repair-deficient prostate cancer tumors can benefit from pembrolizumab (formerly lambrolizumab, Trade Name Keytruda) treatment, given its tissue-agnostic approval by the U.S. Food and Drug Administration (FDA) [26]. This anti-programmed cell death protein 1 (PD-1) checkpoint inhibitor and other immuno-therapeutics might also become a valid option for patients with somatic *BRCA1*/*2* or *ATM* mutations, and for those with *CDK12* biallelic loss [27,28]. In mCRPC, *SPOP* mutations have been suggested to predict the response to abiraterone acetate [29], whereas *PTEN* loss seems to be a predictor of response to Akt (Protein kinase B—PKB) inhibitors [30]. 

Clinical data report that *RB1* loss is associated with worse progression-free survival in patients with mCRPC treated with enzalutamide [31]. In addition, the recent comprehensive analysis of 429 patients with mCRPC has identified that *RB1* alteration was significantly associated with poor survival, and alterations in *RB1* and *TP53* were associated with shorter time on treatment with abiraterone or enzalutamide [32]. Androgen-receptor (AR) amplifications and mutations are mostly the result of hormonal treatment [20], but they can also occur in some castration-naïve patients [33]. The AR alterations confer resistance to ADT, and are associated with a worse prognosis, therefore these patients could be candidates to treatment intensification. The circulating tumor cells (CTC) and the AR splice variant 7 (AR-V7) have been proposed as prognostic and predictive biomarkers in patients with mCRPC treated with abiraterone acetate and enzalutamide [34,35]. However, the prevalence of AR-V7 in untreated mCRPC, and, consequently, in mHSPC, is low [35]. Finally, the role of PSA kinetics, which represent an important criterium for treatment selection in the context of non-metastatic castration-resistant prostate cancer (nmCRPC) [36], remains largely unaddressed in the setting of mHSPC.

## 3. Systemic Treatments in mHSPC 

Several systemic agents are currently available for the treatment of the advanced stages of prostate cancer (Table 2) [37]. In recent years, the role of ADT combined with other treatments or with local therapy has been investigated in many phase 3 trials that led to new therapy approvals for mHSPC (Table 3).

### 3.1. Docetaxel 

Docetaxel is a chemotherapy agent that promotes and stabilizes microtubule assembly, thus inhibiting the mitotic cell division. This was the first drug to demonstrate an improvement in OS in prostate cancer [38]. The benefit of adding docetaxel to lifelong ADT for mHSPC was established by three phase III trials: GETUG-AFU 15, CHAARTED and STAMPEDE (Table 3). In the GETUG-AFU 15 study, which was the first trial of docetaxel in mHSPC [12], 385 patients were randomized to receive ADT plus docetaxel (75 mg/m^2^ intravenously every three weeks up to nine cycles), or ADT alone. Patients in the chemotherapy arm showed improved PSA progression-free survival (22.9 vs. 12.9 months, HR: 0.72 (95% CI, 0.57–0.91); *p* = 0.005, and radiographic progression-free survival (23.5 vs. 15.4 months (HR: 0.75 (95% CI, 0.59–0.94); *p* = 0.015, respectively]. However, these benefits did not translate into improved OS (HR 1.01; 95%CI 0.75–1.36). Four potentially treatment-related deaths and 72 serious adverse events occurred in the experimental arm.

The CHAARTED trial randomized 790 patients to receive ADT + docetaxel (75 mg/m^2^ IV every three weeks up to six cycles) or ADT alone [6]. After a median follow-up of 28.9 months, a statistically significant longer OS was observed in the treatment arm compared to the placebo (57.6 vs. 44 months, HR: 0.61 (95% CI, 0.47–0.80); *p* < 0.001). The median time to biochemical, symptomatic or radiographic progression also favored the combination group (20.2 vs. 11.7 months, HR: 0.61 (95% CI, 0.51–0.72); *p* < 0.001). Among patients who received the combined therapy, 16.7% and 12.6% reported G3 and G4 adverse events, respectively, which were consistent with docetaxel use. At a follow-up of 53.7 months, a longer OS was confirmed in the chemotherapy arm (57.6 vs. 47.2 months, HR: 0.72 (95% CI, 0.59–0.89); *p* = 0.0018). However, the subgroup analysis reported that OS advantage was clear in patients with high-volume disease (*n* = 513) (51.2 vs. 34.4 months, HR: 0.63 (95% CI, 0.50–0.79); *p* < 0.001), but not in those with low tumor burden (*n* = 277) (HR: 1.04 (95% CI, 0.70–1.55); *p* = 0.86) [15]. Although the volume of metastatic disease was not a stratification factor in the GETUG-AFU 15 trial and >75% of patients in this trial were low-volume, post hoc analyses at the 7-year (84 month) follow-up suggested that an interaction might exist between disease volume and docetaxel benefit (HR: 0.78 in patients with high-volume and HR: 1.02 in patients with low-volume) [39].

The STAMPEDE trial is a multi-arm study, with an adaptive design, that evaluates whether the addition of various treatments at the time of ADT initiation improves OS for high risk, locally advanced or mHSPC patients [40]. In the cohort of 593 patients treated with ADT + docetaxel (75 mg/m^2^ IV every three weeks) plus prednisone 10 mg daily (arm C) a significant improvement was observed in both median OS (81 vs. 71 months, HR: 0.78 (95% CI, 0.66–0.93), *p* = 0.006) and failure-free survival (37 vs. 20 months, HR: 0.61 (95% CI, 0.53–0.70) *p* < 0.001) compared to 1184 patients who received ADT alone [41]. The OS benefit appeared to be greater in metastatic patients (HR: 0.76, 95% CI 0.62–0.92), that represented 61% of men enrolled in both arms. 

A meta-analysis on the individual data of patients who were included in these three trials (GETUG-AFU-15, CHAARTED and STAMPEDE) confirmed the OS benefit obtained with the combination of docetaxel plus ADT in men with mHSPC [42]. The combined patient data from these trials showed a 23% reduction in the risk of death (HR: 0.77 (95% CI, 0.68–0.87); *p* < 0.0001), which translated to an absolute improvement in 4-year survival of 9% (95% CI, 5–14). A 36% reduction in the risk of progression was also reported (HR: 0.64 (95% CI, 0.58–0.70); *p* < 0.0001), with a 16% (95% CI, 12–19) reduction in absolute 4-year failure rates.

### 3.2. Abiraterone Acetate 

Abiraterone acetate is an inhibitor of extragonadal androgen biosynthesis that was initially approved for the treatment of mCRPC [43,44]. The addition of abiraterone to ADT has demonstrated to improve OS in two phase III trials, LATITUDE and STAMPEDE (Table 3) [44,45]. Both studies randomized participants to ADT alone, or in combination with abiraterone 1000 mg plus prednisone 5 mg daily, until disease progression or unacceptable toxicity. 

The LATITUDE trial randomized 1,199 patients with newly diagnosed high-risk mHSPC (Table 1). At the interim analysis (30.4 months follow-up), a 38% reduction in the risk of death was observed in patients treated with abiraterone, compared to those who received the placebo (HR: 0.62 (95% CI, 0.51–0.76); *p* < 0.001). Updated data after cross-over and about 2-year additional follow-up confirmed this robust survival benefit (HR: 0.66 (95% CI, 0.56–0.78); *p* < 0.0001) [45]. A 53% reduction in the risk of radiographic progression was also reported with abiraterone (HR: 0.47 (95% CI, 0.39–0.55); *p* < 0.001). Significantly better outcomes in all secondary endpoints were observed in the abiraterone group, including the time until pain progression, next subsequent therapy for prostate cancer, initiation of chemotherapy and PSA progression (*p* < 0.001 for all comparisons), along with the next symptomatic skeletal events (*p* = 0.009). Consistently with the profile of this drug, the most common grade 3–4 adverse events in the intervention arm were hypertension (21% vs. 10% in the placebo) and hypokalemia (12% vs. 2% in the placebo).

In the STAMPEDE trial, 1917 patients with HSPC (52% metastatic) were randomized to receive ADT plus abiraterone acetate 1000 mg daily and prednisolone 5 mg daily (arm G) or ADT alone until progression [44]. Patients treated with abiraterone showed a 37% reduction in the risk of death (HR: 0.63 (95% CI, 0.52–0.76); *p* < 0.001), with an HR of 0.61 (95% CI, 0.49–0.75] in patients with metastatic disease. In addition, a significant reduction in the risk of treatment failure was observed in patients who received abiraterone (HR: 0.29 (95% CI, 0.25–0.34); *p* < 0.001). Grade 3 to 5 adverse events occurred in 47% of patients in the combination group (with nine grade 5 events) and in 33% of the patients in the ADT-alone group (with three grade 5 events). Notably, the retrospective subgroup analysis of OS in 901 metastatic patients included in this cohort did not reveal any interaction among subgroups after stratification according to patients’ risk (LATITUDE criteria) or disease volume (CHAARTED criteria) (Table 1); however, the number of low-risk patients to treat in order to observe the OS benefit was four times greater compared to that of high-risk [46].

### 3.3. Enzalutamide

Enzalutamide is a new generation antiandrogen that is approved for the treatment of mCRPC [47,48,49]. The benefit of adding enzalutamide to ADT for the treatment of mHSPC patients has been established by two phase III studies, ARCHES and ENZAMET (Table 3) [50,51].

The ARCHES trial randomized 1150 men, who were stratified by disease volume and prior docetaxel therapy, to receive ADT plus enzalutamide 160 mg daily, or ADT plus our placebo [50]. At the interim analysis (median follow-up of 14.4 months), the primary endpoint of improved radiographic progression-free survival (rPFS) was met (HR: 0.39 (95% CI, 0.30–0.50); *p* < 0.001). The benefit on rPFS was consistent across all pre-specified subgroups, including disease volume and prior docetaxel chemotherapy. Significant improvements in secondary outcome measures, such as time to the initiation of a new antineoplastic therapy, time to PSA progression, PSA undetectable rate and objective response rate, were also observed. At the time of this interim analyses, the data on OS were reported as immature.

The phase III ENZAMET trial investigated the efficacy of enzalutamide in 1125 patients with mHSPC, who were randomized to receive medical or surgical castration, plus either enzalutamide 160 mg daily, or conventional non-steroidal antiandrogen until disease progression or prohibitive toxicity [51]. Different from the ARCHES trial, the primary endpoint of ENZAMET was OS. 

This was the first study to examine the use of an ARSi with or without concurrent docetaxel, and 45% of patients enrolled were planned to receive docetaxel. Men were stratified according to disease volume as per CHAARTED criteria, anti-resorptive therapy, comorbidities, planned early docetaxel use and study site. After a median follow-up of 33 months, patients treated with enzalutamide plus ADT showed longer survival compared to those treated with conventional non-steroidal antiandrogen plus ADT (HR: 0.66 (5% CI, 0.51–0.86); *p* = 0.0016). At 3 years, 79% and 72% were still alive in the experimental and control arms, respectively. Pre-specified subgroup analyses suggested that the benefit of enzalutamide was less clear in patients with high-volume disease (HR: 0.74 (95% CI, 0.55–1.01)) and in those planned to receive early docetaxel (HR: 0.91 (95% CI, 0.62–1.35)). However, this trial was neither designed nor powered to reliably analyze the results in these subgroups. Serious adverse events (regardless of attribution) within 30 days of study treatment occurred in 42% and 34% of patients enrolled in the experimental and control arms, respectively. 

### 3.4. Apalutamide 

Apalutamide is a selective androgen-receptor (AR) antagonist that is approved for the treatment of nmCRPC, based on the SPARTAN study [36]. The phase III TITAN trial demonstrated that an addition of apalutamide to lifelong ADT also improved OS in mHSPC (Table 3) [52]. The study randomized 1052 participants to ADT alone or in combination with apalutamide 240 mg per day. Participants were stratified by Gleason score, region and prior docetaxel use. In addition, rPFS and OS were co-primary endpoints. Secondary endpoints included time to pain progression, time to skeletal-related event, time to chronic opioid use and time to initiation of cytotoxic chemotherapy. Most participants presented de novo mHSPC, and 16% of patients had received prior treatment for localized disease and were enrolled at metastatic relapse. The trial was amended to allow eligibility of patients who had received docetaxel for mHSPC, but only 11% of patients enrolled had previous exposure to chemotherapy. Most patients had high-volume disease (63%) as per CHAARTED criteria (Table 1). At median follow-up of 22.6 months, apalutamide significantly improved rPFS (HR: 0.48 (95% CI, 0.39-0.60); *p* < 0.0001), with a 52% reduction in the risk of death or radiographic progression. Apalutamide also significantly improved OS (HR: 0.67 (95% CI, 0.51–0.89); *p* = 0.0053), with a 33% reduction in risk of death, and no significant differences according to disease volume. In addition, time to initiation of cytotoxic chemotherapy was significantly longer with apalutamide (HR, 0.39; 95% CI, 0.27–0.56; *p* < 0.0001). Due to the limited number of patients that received apalutamide after docetaxel, the benefit in survival of this sequential strategy remains unclear. Increased risks of all grade rash (27.1% vs. 8.5%), pruritus (10.7% vs. 4.6%), hot flushes (22.7% vs. 16.3%), hypothyroidism (6.5% vs. 1.1%) and fractures (6.3% vs. 4.6%) were reported in the intervention arm. However, rates of grade 3/4 adverse events (AEs) were similar between the groups, and discontinuations due to AEs were 8% and 5% for apalutamide and placebo arms, respectively. 

### 3.5. Other Potential Systemic Treatments for mHSPC

#### 3.5.1. Bone-Targeting Agents

Bisphosphonates and denosumab have failed to improve OS in mCRPC patients. However, both have shown to delay or prevent skeletal-related events (SREs) in patients with mCRPC [53,54], and have therefore been approved for this setting. Bone metastases are found in approximately 90–95% of patients with mHSPC [16,55], but, different from mCRPC, there is no evidence to recommend bone protective agents in mHSPC. 

The benefit of adding zoledronic acid (4 mg intravenously every four weeks) to ADT in the mHSPC scenario was analyzed by the CALGB 90202 study that randomized 645 men with HSPC and bone metastases to receive zolendronate or placebo [56]. The primary endpoint of improved time to first SRE was not met (31.9 vs. 29.8 months in the treatment and placebo groups, respectively) and the study was prematurely discontinued. The STAMPEDE trial also analyzed the benefit of adding zoledronic acid to ADT and to docetaxel plus ADT in patients with high-risk, locally advanced or mHSPC [41]. Addition of zoledronic to ADT alone or to ADT plus docetaxel did not show any evidence of OS advantage (HR 0.94 (95% CI, 0.79–1.11); *p* = 0.45; HR 1.06 (95% CI, 0.86–1.30); *p* = 0.59, respectively). No benefit in time to first SRE was either observed in those who received ADT + zoledronic acid, compared to those who received ADT alone (HR 0.89 (95% CI, 0.73–1.07); *p* = 0.2). In the PR05 trial, 311 men with mHSPC who were starting or responding to first-line hormone therapy were randomly assigned to receive oral sodium clodronate (2,080 mg/day), or a placebo [57]. The primary endpoint of improved bone PFS was not met, although a non-statistically significant trend for better bone PFS was observed in the group treated with clodronate (HR: 0.79 (95% CI, 0.61–1.02); *p* = 0.06). The long-term follow-up results also suggested that patients treated with clodronate had longer OS (HR: 0.77 (95% CI, 0.60–0.98), *p* = 0.032) compared to those in the control group [58]. To date, no trial has evaluated the role of denosumab in mHSPC. 

#### 3.5.2. Darolutamide

Darolutamide is a novel nonsteroidal androgen-receptor antagonist that has shown to prolong OS in nmCRPC [59]. The phase III ARASENS trial evaluates the safety and efficacy of darolutamide in addition to ADT and chemotherapy in mHSPC [60]. Currently, 1303 patients with mHSPC are randomized in a 1:1 ratio to receive 300 mg of darolutamide/placebo twice daily, in addition to ADT and docetaxel for six cycles. The primary objective is to show superior OS with darolutamide versus placebo, both with ADT + docetaxel. Secondary endpoints include time to CRPC, initiation of subsequent anticancer therapy, symptomatic skeletal event-free survival (SSE-FS), time to first SRE, initiation of opioid use, pain progression and worsening of physical symptoms. The estimated primary completion date of this study is August 2022.

#### 3.5.3. Other Systemic Treatment Strategies Under Evaluation 

The STAMPEDE study is a multi-arm, multi-stage phase III study designed to test whether the addition of various treatments at the time of ADT initiation improves OS. This trial includes patients with both M0 and M1 HSPC, and men are randomized to ADT alone or in combination with various therapies. The arm J of the STAMPEDE trial is currently investigating whether the addition of enzalutamide plus abiraterone to ADT improves OS over ADT alone. Overall, 1,800 patients starting long-term hormone therapy will be randomized 1:1 to the control arm A (currently ADT with additional prostate RT for N0M0 patients), or to the research arm J. Two intermediate analyses on failure-free survival are planned, and mature dates on OS data are expected for 2020 [61]. An indirect comparison of the relative efficacy of adding enzalutamide to abiraterone will be possible through the published results of the separate STAMPEDE “abiraterone comparison” (arm G) [44].

In the arm K of STAMPEDE, approximately 1700 non-diabetic patients with mHSPC are being randomized to investigate the role of metformin in mitigating the debilitating effects of prolonged ADT [62]. Standard of care plus metformin will be compared in terms of OS benefit to the current standard of care (control arm A). Finally, arm L will analyze the clinical efficacy and side effect profile of transdermal estradiol versus standard ADT for men with locally advanced or mHSPC [40]. Transdermal estradiol should avoid the toxicities associated with estradiol deficiency observed during long-term ADT [63]. Approximately 500 patients will be included within a meta-analysis with the PATCH trial, which will include around 2000 patients overall. 

Co-primary endpoints are progression-free survival and OS. In the phase III PEACE I trial, 1173 patients are randomized in four arms that compare ADT + docetaxel vs. ADT + docetaxel + abiraterone acetate vs. ADT + docetaxel + radiotherapy to primary tumor vs. ADT + docetaxel + abiraterone acetate + radiotherapy to primary tumor [64]. Patients are stratified according to performance status, disease extent (lymph nodes only vs. bone vs. presence of visceral metastases). The results of this trial are expected to provide important information about the best treatment strategy for mHSPC.

As mentioned in paragraph 2.3, the role of other specific treatments, such as PARP inhibitors, Akt inhibitors or immunotherapy is currently uncertain in the mHSPC setting, and a biomarker-driven selection of patients might provide new treatment options for these patients in the next future [65]. Similarly, the prostate-specific membrane antigen (PSMA) targeted-therapy has shown promising results in phase II trials in mCRPC, but its role in mHSPC is still unaddressed [66].

## 4. Treatment of Primary Tumor in mHSPC

### 4.1. Radiotherapy to Primary Tumor

The role of the treatment of the primary tumor is recognized in metastatic renal cancer [67]. The HORRAD and STAMPEDE trials addressed the efficacy of this strategy in mHSPC (Table 3) [55,68]. In the HORRAD trial, 432 men with newly-diagnosed HSPC, PSA > 20 ng/mL and bone metastases, were randomized to receive ADT with or without prostate radiotherapy [68]. Median time to PSA progression was longer in the radiotherapy group compared to ADT alone (15 vs. 12 months, HR: 0.78 (95% CI, 0.63–0.97); *p* = 0.02). Data on OS resulted inconclusive (HR: 0.90 (95% CI, 0.70–1.14)), but this trial raised the possibility that survival might be improved in a subgroup of patients with fewer than five bone metastases (HR: 0.68 (95% CI, 0.42–1.10)). 

In the STAMPEDE trial, a cohort of 2,061 patients with newly diagnosed mHSPC were randomized to receive the standard of care or standard of care plus radiotherapy to the primary tumor (arm H) [55]. Upfront docetaxel was allowed, but only 18% of patients received chemotherapy in addition to ADT. Radiotherapy to the primary was started within 3–4 weeks after the last docetaxel dose (55Gy in 20 fractions over four weeks or 36Gy in 6 fractions over six weeks). Radiotherapy improved failure-free survival (HR: 0.76 (95% CI, 0.68–0.84); *p* < 0.0001), but not OS (HR: 0.92 (95% CI, 0.80–1.06); *p* = 0.266). However, in the pre-specified analysis, the low-volume subgroup, as per CHAARTED criteria (Table 1), had significant benefit in both failure-free (HR 0.59, 95% CI 0.49–0.72) and overall survival (HR 0.68, 95% CI 0.52–0.90).

### 4.2. Cytoreductive Prostatectomy 

The role of cytoreductive prostatectomy in mHSPC has not been properly addressed. Large retrospective data based on US SEER-Medicare and U.S. National Cancer Data Base suggest that radical prostatectomy might confer a significant survival advantage to patients with de novo mHSPC [69,70,71]. Patients with lower tumor grade, local stage, metastatic substage and good general conditions seem to benefit the most from this approach [70,71]. In a prospective case-control study, Steuber and colleagues compared the outcomes of 43 patients with low-volume bone metastases from prostate cancer (1–3 lesions) undergoing cytoreductive prostatectomy, and 40 patients receiving systemic therapy [72]. No significant differences in castration resistant-free survival or OS were found between arms, but patients treated with prostatectomy had less locoregional complications. In the prospective LoMP trial, 17 asymptomatic patients with mHSPC underwent surgery, and 29 patients ineligible or unwilling to undergo radical prostatectomy received standard of care [73]. Patients treated with radical prostatectomy showed better OS, PSA response and longer time to ADT failure compared to those included in the control arm. However, patients included in the experimental arm were younger, had lower PSA at diagnosis, and showed less extensive local and metastatic disease. 

Therefore, a significant selection bias might affect the reliability of the data that are currently available, and further prospective studies are warranted to determine the potential benefit of cytoreductive prostatectomy in patients with mHSPC.

## 5. Oligometastatic Disease

A multimodal approach should be considered in patients with oligometastatic HSPC. However, several concerns remain with respect to the definition of oligometastatic disease [74], and most studies used a definition of ≤3 metastases in the recurrent or de novo HSPC setting [75]. The STOMP trial was a multicenter, randomized, phase II study that enrolled 62 patients with asymptomatic mHSPC who had biochemical recurrence after primary treatment and three or fewer extra-cranial metastatic lesions [76]. Patients were randomized to either surveillance or metastasis-directed therapy (MDT) of all detected lesions (surgery or stereotactic body radiotherapy). At a median follow-up time of three years, the median ADT-free survival was 13 months (80% CI, 12–17) for the surveillance group and 21 months (80% CI, 14–29) for the MDT group (HR: 0.60 (80% CI, 0.40–0.90); *p* = 0.11). The major limitation, beyond the small sample size, is that ADT-free survival was the primary endpoint, and the control arm strategy was surveillance instead of ADT [77]. However, this was the first randomized trial to suggest that MDT might be useful in oligometastatic HSPC. The ORIOLE study has a design similar to the STOMP trial, and the preliminary analysis on 24 men seems to confirm the results of this STOMP trial, showing that MDT might be safe, and delay disease progression [78]. 

## 6. Choosing the Right Treatment for the Right Patient

The lack of direct comparisons among all of the new therapeutic strategies for mHSPC represents a challenge for scientists and clinicians. A summary of the phase 3 randomized controlled trials in mHSPC is shown in Table 3, and potential decision-making factors when selecting the treatment for mHSPC are described in Table 4.

### 6.1. Comparing Patient’s Populations and Survival Benefit

In terms of efficacy, the general improvement in OS is quite similar among the different phase 3 trials and it is affected by the heterogeneity of populations and by different patients’ baseline characteristics (Table 3). A network meta-analysis tried to assess the optimal systemic treatment in mHSPC between docetaxel and abiraterone acetate, including aggregate data from STAMPEDE, GETUG-AFU 15, CHAARTED and LATITUDE trials [79]. The results suggested that abiraterone acetate plus prednisone and ADT had the highest probability of being the most effective treatment both for OS (94% probability) and failure-free survival (100% probability), with docetaxel plus ADT the second most effective treatment. However, the authors remarked that it was not clear to what extent, and whether this was due to a true increased benefit with abiraterone or to the variable features of the individual trials. The direct, randomized, comparative analysis of the STAMPEDE trial did not reveal any difference in overall and cancer-specific survival between abiraterone and docetaxel [80]. In this analysis, worst adverse events were similar, and included different toxicities that were consistent with the known properties of the drugs. As previously mentioned, the disease volume and/or risk may be important criteria for treatment choice. In the STAMPEDE trial, survival outcomes were similar according to disease volume or risk after treatment with abiraterone acetate [81]. However, the role of docetaxel in low-volume disease is uncertain [15,39], and the interpretation of additional risk factors may be relevant to select patients likely to benefit from the addition of chemotherapy to ADT (Table 4). Conversely, the survival benefit of radiotherapy to the primary tumor is not demonstrated in high-volume patients [55]. Likewise, the subgroup analysis of the ENZAMET trial suggests that the benefit of enzalutamide might be reduced in patients with high-volume disease, but it is not clear if prior docetaxel use affects this observation [51].

The time of metastatic presentation affects patients’ prognosis [17,18], and the patients with metastatic recurrence after radical treatment, who are expected to show better outcomes than those with de novo metastatic disease, were not adequately represented in the majority of the phase 3 trials in the mHSPC setting (Table 3). The LATITUDE trial only enrolled patients with de novo mHSPC, and only 4% of patients included in the STAMPEDE “abiraterone” arm were relapsing after radical treatment. Therefore, the benefit of adding abiraterone to ADT in the last patients’ population is uncertain. High-volume patients with prior local therapy included in the CHAARTED trial showed a trend that was similar to patients with high-volume de novo disease [15], however further studies should specifically investigate the role of adding chemotherapy and ARSi to ADT in patients with recurrence after local therapy. Regarding patients with de novo disease, a network meta-analysis tried to compare the efficacy of abiraterone acetate versus docetaxel according to disease volume and risk in patients with newly-diagnosed mHSPC included in GETUG-AFU 15, CHAARTED and LATITUDE trials [82]. An 8% relative reduction in mortality was observed for newly diagnosed high-risk patients treated with abiraterone acetate than those treated with docetaxel + ADT (HR 0.92 (95% CI, 0.69–1.23)), with the Bayesian probability of abiraterone acetate being the better treatment found to be 71.8%. In this patients’ population, abiraterone acetate was also associated with a 24% reduction in the risk of radiographic progression or death compared with docetaxel (HR 0.76 (95% CI, 0.53, 1.10)), and the Bayesian probability of abiraterone acetate being the better treatment was 92.9%. The comparison of other secondary endpoints beyond OS is challenging. For example, the different definitions of progression-free survival among the randomized trials make the results not comparable. Data on progression-free survival 2 are largely unknown, and their availability could allow to better understand the most appropriate treatment sequence for patients with mHSPC.

Finally, in most of the trials involving mHSPC patients, the metastatic disease was assessed by computed tomography and bone scan. Only the studies conducted in the oligometastatic setting have used choline or prostate-specific membrane antigen positron emission tomography (PET) to define the disease extent [76]. These imaging techniques have different sensitivities, and patients who are considered to be metastatic by PET may not be by TC or bone scan, thus making difficult any future comparisons between these studies. 

### 6.2. Factors that May Influence Treatment Decision

The drug mechanism of action, the route of administration, the duration of treatment, the impact on quality of life and the toxicity profile are important factors to consider when selecting a therapy for a particular patient, as they are quite different among the various strategies (Table 2 and Table 4). Docetaxel has a major incidence of myelo-suppression with potential neutropenia, fatigue and neurotoxicity. Abiraterone is associated with mineralocorticoid-associated side effects including hypertension, hypokalemia and hepatic toxicity. Enzalutamide frequently causes fatigue, hypertension and falls. Apalutamide is associated with increased risk of rash, pruritus, hot flushes, hypothyroidism and fractures. Radiotherapy to the primary tumor can cause acute and late bladder and bowel toxic effects that can remarkably affect the quality of life (QoL). 

Patient preferences and comorbidities need to be considered before starting treatments. Oral agents are expected to provide better acceptance and to avoid the toxicities of chemotherapy and radiotherapy. However, these last strategies have the advantage of shorter treatment duration, and some patients may will to avoid the use of continuous oral therapies. Use of abiraterone acetate might be challenging in diabetic or osteoporotic patients, given the concurrent use of steroids. Patients with a history of or risk factors for seizures were excluded from controlled clinical studies with enzalutamide, although the UPWARD study has not observed an increased incidence of seizures in enzalutamide-treated patients with a personal history of seizure or other predisposing factors [83]. It is also important to consider potential interactions between drugs, as prostate cancer patients are often on multiple medications for concurrent comorbidities.

In terms of QoL, patients treated with chemo-hormonal therapy in the CHAARTED trial reported a significant decline in the Functional Assessment of Cancer Therapy-Prostate (FACT-P) compared to those who received ADT alone at three months [84]. However, FACT-P was recovered at 12 months, and did not significantly differ with baseline FACT-P. The QoL analysis of the LATITUDE trial, as assessed by the Brief Pain Inventory (BPI), FACT-P and EuroQoL (5-Level EQ-5D version), reported a significant clinical benefit in several QoL endpoints in patients treated with abiraterone acetate, compared to those who received ADT alone [85]. The meta-analysis performed by Feyerabend et al. compared QoL in newly diagnosed mHSPC patients treated with docetaxel or abiraterone in addition to ADT [82]. Abiraterone plus ADT resulted in more favorable outcomes in terms of BPI and FACT-P in both high-risk and high-risk/high-volume populations. The QoL analysis from the ARCHES trial did not show any significant difference between enzalutamide plus ADT versus ADT alone for time to deterioration in FACT-P [50]. Enzalutamide plus ADT significantly delayed time to pain progression for worst pain (HR 0.82 (95% CI, 0.69–0.98); *p* = 0.03) and pain severity (HR 0.79 (95%CI, 0.65–0.97); *p* = 0.02) versus ADT alone. In the TITAN trial, analysis of change from baseline in the FACT-P score with the use of a mixed-effect repeated-measures model showed that health-related QoL was maintained with apalutamide, with no substantial between-group difference [52].

In terms of costs, docetaxel in combination with ADT is likely to be the most cost-effective treatment option for patients with mHSPC [86]. Docetaxel is administered every 21 days for six cycles at an approximate cost of $550 per cycle, whereas novel ARSi are prescribed as a daily dosing schedule until the time of progression at an approximate cost that exceeds $7000 per month (abiraterone acetate is currently less expensive with the availability of generic formulations) [86]. To administer lower dosages of ARSi might be an opportunity to reduce toxicities and costs, but phase 3 non-inferiority trials are still needed [87]. 

In addition, it is currently unclear whether all patients should be treated with potentially toxic combination therapy or with ADT alone, and whether first-generation antiandrogens, such as bicalutamide, should be definitely abandoned. 

## 7. Conclusions

Management of mHSPC has completely evolved during the last years. Both chemotherapy and ARSi demonstrated a significant survival benefit when combined to ADT compared to ADT alone. Radiotherapy to the primary tumor is a new standard of care in low-volume mHSPC, and further studies are needed to assess the role of cytoreductive prostatectomy. Despite the fact that many treatment options are currently implemented in the international guidelines for mHSPC patients, no data on the optimal treatment sequence are available. Treatment choice is based upon indirect comparisons of randomized trials and on the specific characteristics of each patient. New biomarkers are therefore warranted to improve patients’ selection.

## Figures and Tables

**Table 1 cancers-11-01355-t001:** High-risk patients according to the CHAARTED and LATITUDE criteria.

**CHAARTED criteria** [6]	Visceral Metastases AND/OR ≥4 Bone Lesions (with ≥1 beyond the Vertebral Bodies and Pelvis)
**LATITUDE criteria** [16]	≥2 high-risk features ≥3 bone metastases visceral metastases Gleason ≥ 8

**Table 2 cancers-11-01355-t002:** Systemic agents that are currently approved for the treatment of prostate cancer.

Drug	Mechanism of Action	Administration
***Agents with approval for the treatment of mHSPC***		
LH-RH agonists and antagonists	Inhibition of LH and FSH release	IM/SC every 30, 90 or 180 days
alone or in combination with:		
1st generation antiandrogens - Bicalutamide - Nilutamide - Flutamide	1st generation nonsteroidal antiandrogens	Continuous oral - 50 mg daily - 150 mg daily - 750 mg daily
Docetaxel	Microtubule assembly inhibitor	75 mg/m^2^ IV 3-weekly for six cycles
Abiraterone acetate	Androgen biosynthesis inhibitor	Continuous oral 1000 mg daily with prednisone 10 mg daily
Enzalutamide	2nd generation antiandrogen	Continuous oral 160 mg daily
Apalutamide	2nd generation antiandrogen	Continuous oral 240 mg daily
***Agents with approval for other prostate cancer settings***		
Cabazitaxel	Microtubule assembly inhibitor	IV 3-weekly up to 10 cycles 20/25 mg/m^2^
Darolutamide	2nd generation antiandrogen	Continuous oral 1200 mg daily
Radium 223 dichloride	α-emitting radionuclide	IV 4-weekly for six doses 55 kBq/Kg
Sipuleucel-T	Cancer vaccine	IV 2-weekly for three infusions
Bone-targeting agents - Bisphosphonates - Denosumab	- Osteoclasts inhibition - RANKL inhibition	- IV 4/12-weekly - SC 4-weekly
Mitoxantrone	Topoisomerase II inhibitor	IV 3-weekly 12 mg/m^2^
Estramustine phosphate	Alkylating and estrogenic activity	Continuous oral 14 mg/kg

Abbreviations: FSH: Follicle-stimulating hormone; IM: Intramuscular; IV: Intravenous; LH: Luteinizing hormone; LH-RH: Luteinizing hormone-releasing hormone; RANKL: Receptor activator of nuclear factor kappa-Β ligand; SC: Subcutaneous; mHSPC: Metastatic hormone-sensitive prostate cancer.

**Table 3 cancers-11-01355-t003:** Summary of phase 3 randomized controlled trials in metastatic hormone-sensitive prostate cancer (mHSPC).

STUDY CHARACTERISTICS		GETUG-AFU 15	CHAARTED	STAMPEDE	LATITUDE	STAMPEDE	ARCHES	ENZAMET	TITAN	HORRAD	STAMPEDE
		Docetaxel	Docetaxel	Docetaxel	Abiraterone	Abiraterone	Enzalutamide	Enzalutamide	Apalutamide	Radiotherapy	Radiotherapy
**STUDY DESIGN**	**Inclusion criteria**	mHSPC	mHSPC	High-risk, locally-advanced or mHSPC	Newly diagnosed high-risk mHSPC	High-risk, locally-advanced or mHSPC	mHSPC	mHSPC	mHSPC	Untreated bone mHSPC	Newly diagnosed mHSPC
	**Stratification factors**	Treatment for primary tumor, systemic therapy for PSA relapse, Glass risk groups	Age, PS, planned use of CAB or bone-agents, duration of prior ADT, HV/LV	Hospital, age, M1, N1, PS, planned ADT or RT, use of aspirin or NSAIDs	Measurable visceral disease, PS	Hospital, age, M1, N1, PS, planned ADT or RT, use of aspirin or NSAIDs	HV/LV, prior docetaxel	HV/LV, planned docetaxel, planned bone-agents, comorbidity score, site	Gleason score, geographic region, prior docetaxel use	None	Hospital, age N1, PS, planned ADT, use of aspirin or NSAIDs, planned docetaxel
	**Primary endpoint**	OS	OS	OS	OS	OS	rPFS	OS	OS and rPFS	OS	OS
**POPULATION**	**Patients (*n* = exper/placebo)**	192/193	397/393	592/1184	597/602	960/957	574/576	563/562	525/527	216/216	1032/1029
	**Age (years)**	63/64	64/63	65/65	68/67	67/67	70/70	69/69	69/68	67/67	68/68
	**Gleason > 7 (%)**	55/59	61/62	74/68	98/97	74/75	67/65	60/57	67/68	65/66	82/83
	**Prior radical treatment (%)**	33/24	27/27	3/3 in M1	0/0	4/3 in M1	25/28	42/42	18/15	0/0	0/0
	M1 (%)	100/100	100/100	62/61	100/100	52/53	93/92	100/100	100/100	100/100	100/100
	**High volume (%)**	48/47	66/64	NR	82/78	55.4% in M1	62/65	52/53	62/64	NR	57/58
	**Median PSA (ng/mL)**	27/26	51/52	70/67	NR	51/56	5/5	NR	6/4	125/149	97/98
	**Median FU (mo)**	84 ^†^	54 ^†^	43	52 ^†^	40	14	34	23	47	37
**EFFICACY**	**OS**	62.1/48.6 mo HR: 0.88 (0.68–1.14)	57.6/47.2 mo HR: 0.72 (0.59–0.89)	HR: 0.78 (0.66–0.93) *M1:* HR: 0.76 (0.62–0.92)	53.3/36.5 mo HR: 0.66 (0.56–0.78)	HR: 0.63 (0.52-0.76) *M1*: HR: 0.61 (0.49–0.51)	NE/NE HR: 0.81 (0.53–1.25) *	NE/NE HR: 0.67 (0.52–0.86)	NE/NE HR: 0.67 (0.51–0.89)	45/43 mo HR: 0.90 (0.70–1.14)	42.5/41.6 mo HR: 0.92 (0.80–1.06)
	**OS HV** ^♦^	39.8/35.1 mo HR: 0.78 (0.56–1.09)	51.2/34.4 mo HR: 0.63 (0.50–0.79)	NR	49.7/33.3 mo HR: 0.62 (0.52–0.74)	HR: 0.60 (0.46–0.78)	NR	HR: 0.80 (0.59–1.07)	NE/NE HR: 0.68 (0.50–0.92)	*≥**5 bone M1* HR: 1.06 (0.80–1.39)	37.6/38.8 mo HR: 1.07 (0.90–1.28)
	**OS LV** ^♦^	NE/83.4 mo HR: 1.02 (0.67–1.55)	63.5/NE HR: 1.04 (0.70–1.55)	NR	NE/NE HR: 0.72 (0.47–1.10)	HR: 0.64 (0.42-0.97)	NR	HR: 0.43 (0.26–0.72)	NE/NE HR: 0.67 (0.34–1.32)	*<5 bone M1* HR: 0.68 (0.42–1.10)	49.1/45.4 mo HR: 0.68 (0.52–0.90)
**SAFETY**	**Most frequent ≥3 AEs in experimental arm**	Neutropenia (32%), febrile neutropenia (7%), fatigue (7%)	Neutropenia (12%), febrile neutropenia (6%), fatigue (4%)	Neutropenia (12%), febrile neutropenia (15%), general (7%) and GI disorder (8%)	Hypertension (21%), hypokalemia (12%), ALT (5%) AST (4%) increase	Hypertension (5%), CV disorder (10%), hepatic disorder (7%)	Hypertension (3%)	Hypertension (8%), neutropenia (6%), fatigue (6%), syncope (4%)	Rash (6%), asthenia (2%)	NR	Overall 5%
**Ref**		[12,39]	[6,15]	[41]	[16,45]	[44]	[50]	[51]	[52]	[68]	[55]

Abbreviations: ADT: Androgen-deprivation therapy; AEs: Adverse events; CAB: Complete androgen-blockade; FU: Follow-up; HR: Hazard ratio; HV: High-volume disease; LV: Low-volume disease; M1: Metastatic disease; mHSPC: Metastatic hormone-sensitive prostate cancer; N1: Node-positive disease; NE: Not estimable; NR: Not reported; NSAIDs: Nonsteroidal anti-inflammatory drugs; OS: Overall survival; PS: Performance status; PSA: Prostate-specific antigen. ^†^ Results are updated to the last survival analysis that was available. * OS was not the primary endpoint of the ARCHES trial. ♦ Subgroup analyses according to CHAARTED criteria (pre-specified or exploratory).

**Table 4 cancers-11-01355-t004:** Potential decision-making factors in mHSPC.

**Benefit in trial endpoints**	Overall Survival and Cancer-Specific Survival
	Time to Castration-Resistance, PSA Progression-Free Survival (PSA-PFS), Radiographic Progression-Free Survival (rPFS) and Progression-Free Survival (PFS)
	Quality of Life (QoL)
**Disease characteristics**	Disease volume and risk
	Gleason score
	Presence of visceral metastasis
	Localization of bone metastasis (appendicular or axial skeleton)
	Timing of metastatic disease (de novo or recurrence)
	Oligometastatic disease
**Patient characteristics**	Age
	Performance status
	Concurrent comorbidities
	Preference for oral or IV agent
	Pain score
**Specific alterations**	Alterations in DNA repair pathway (BRCA1/2, PALB2, ATM loss, CDK12 loss)
	RB1 loss
	AR aberrations (AR gain, AR-V7 expression)
	PTEN loss
	SPOP mutations
	Mismatch repair defects (MMR)
**Laboratory values**	Phosphatase alkaline (ALP)
	Lactate dehydrogenase (LDH)
	CTC count
	PSA kinetics
**Drug characteristics**	Specific side effects
	Duration of treatment
	Mechanism of action
	Costs
	Route of administration

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
