# Peer review of "Current Treatment Options for Metastatic Hormone-Sensitive Prostate Cancer"

_cancers, 2019, doi:10.3390/cancers11091355_

Round 1

Reviewer 1 Report

Overall the reviews is well written and could be useful to the oncologists who treat the PCa, even if a deeper description of of the bio-cellular mechanisms of both the PCa and the drugs used would give another thickness to the review and make it perfect for “Cancers”.

Clinical aspects of PCa seems to be treated in a complete way, with the citation of all the drugs currently used for the PCa.  However, if in my opinion a table inserted at the beginning of section 4 which includes the different drugs, their cellular targets and the stage of tumor at which they are applied would help to improve the clarity and readability of the review.

In my opinion the Introduction section should be deepened, giving details on the different stages of prostate cancer. In addition Introduction and Natural History sections can be merged.

Authors should mention that another classification exists in addition to the Gleason score: “Since 2016, a new classification has been proposed and progressively introduced, called Grade Group (GG), which distinguishes tumors in 5 groups. This new classification has the advantage of greater simplicity and immediacy (the lowest grade is 1) and correlates better with prognosis. In the transition phase from one to the other, both are reported in the reports. "

Section 3 (prognostic factors) can be merged with the Biomarkers section 7.4 (it clashes in this position)

Missing and incomplete references:

I can't find the reference 32 the references 34-41-48-65-80 are incomplete (no volume and pages).

Author Response

We thank the reviewer for his/her positive comments and suggestions. Following his/her recommendation, we have merged the introduction and natural history sections and amended references 32, 34, 41, 48, 65 and 80. We inform that bibliography has been updated with additional references.

As suggested, we have put a reference about new Gleason score classification and we have also moved the biomarkers section after the prognostic factors section. A new table about the different systemic agents used in prostate cancer with their mechanism of action has been put in section 3. We agree that a through description of the bio-cellular mechanisms will make our work more complete, however this is beyond the scope of this review and has extensively been discussed elsewhere. In addition, according to the policy of Cancers, the maximum number of words is fixed to 6.000 and the addition of more paragraphs would require a significant extension in words limit.

Reviewer 2 Report

This manuscript is a summary of the outcomes of phase 3 randomized controlled trials for metastatic hormone-sensitive prostate cancer (mHSPC). 

This summary is scientifically sound and it covers recently published trials such as ENZAMET. However, it is hard to refer this manuscript as a high priority "Review Article" while it it could be considered for considered for publication after the following revisions: 

1-  Add a paragraph about PSMA-targeted therapy and its current status. 

2- Follow the instructions for acronyms 

Author Response

We thank the reviewer for his/her comments. We have accordingly added a paragraph and bibliography reference regarding the potential use of PSMA-targeted therapies for mHSPC. The abbreviations have been inserted according to Cancers Guidelines for Authors.